# Unleashed Actin Assembly in Capping Protein-Deficient B16-F1 Cells Enables Identification of Multiple Factors Contributing to Filopodium Formation

**DOI:** 10.3390/cells12060890

**Published:** 2023-03-14

**Authors:** Jens Ingo Hein, Jonas Scholz, Sarah Körber, Thomas Kaufmann, Jan Faix

**Affiliations:** Institute for Biophysical Chemistry, Hannover Medical School, Carl-Neuberg-Strasse 1, 30625 Hannover, Germany

**Keywords:** filopodia, capping protein, Ena/VASP proteins, myosin-X, mDia2, FMNL2, FMNL3

## Abstract

Background: Filopodia are dynamic, finger-like actin-filament bundles that overcome membrane tension by forces generated through actin polymerization at their tips to allow extension of these structures a few microns beyond the cell periphery. Actin assembly of these protrusions is regulated by accessory proteins including heterodimeric capping protein (CP) or Ena/VASP actin polymerases to either terminate or promote filament growth. Accordingly, the depletion of CP in B16-F1 melanoma cells was previously shown to cause an explosive formation of filopodia. In Ena/VASP-deficient cells, CP depletion appeared to result in ruffling instead of inducing filopodia, implying that Ena/VASP proteins are absolutely essential for filopodia formation. However, this hypothesis was not yet experimentally confirmed. Methods: Here, we used B16-F1 cells and CRISPR/Cas9 technology to eliminate CP either alone or in combination with Ena/VASP or other factors residing at filopodia tips, followed by quantifications of filopodia length and number. Results: Unexpectedly, we find massive formations of filopodia even in the absence of CP and Ena/VASP proteins. Notably, combined inactivation of Ena/VASP, unconventional myosin-X and the formin FMNL3 was required to markedly impair filopodia formation in CP-deficient cells. Conclusions: Taken together, our results reveal that, besides Ena/VASP proteins, numerous other factors contribute to filopodia formation.

## 1. Introduction

The precisely coordinated spatiotemporal control of the assembly and disassembly of actin filaments is a major determinant in a wide range of fundamental cellular processes such as endocytosis, cytokinesis and cell migration [1,2,3,4]. Specific protein assemblies, composed of numerous actin-binding proteins, act in these processes to nucleate, elongate or cap new actin filaments, organize them into complex 3D arrays, and, subsequently, disassemble them to replenish the polymerization-competent pool of monomeric G-actin [5,6]. Migration of cells on flat and rigid substrates is commonly initiated by the protrusion of sheets of cytoplasm, referred to as lamellipodia, which are filled with dendritic actin filament networks and compact filament bundles termed microspikes and filopodia, the structure, dynamics and turnover of which have been extensively characterized [7,8,9,10]. The nucleation of branched actin networks in lamellipodia is driven by the actin-related proteins 2 and 3 (Arp2/3) complex downstream of the WAVE-regulatory complex (WRC) and Rac signaling [11,12,13,14]. The only protein families known so far to actively accelerate the rate of actin filament elongation in the presence of CP (also known as CapZ in muscle) by incorporating actin monomers at the growing barbed ends are formins and Ena/VASP proteins, albeit their modes of action differ considerably [10].

Formins constitute a conserved group of large, multi-domain proteins that promote the nucleation and elongation of unbranched (linear) actin filaments as found in stress fibers (SF) and filopodia [15]. The crescent-shaped FH2 domain homo-dimerizes into a doughnut-like structure [16], nucleates actin filaments and remains continuously associated with the progressively elongating barbed end, thereby effectively preventing termination of filament growth by CP [17]. The FH2 domain is usually preceded by a formin homology 1 (FH1) domain composed of consecutive stretches of poly-L-proline that serve as binding sites for profilin–actin complexes, enabling recruitment and delivery of ATP-loaded actin monomers to the FH2 domain for subsequent incorporation into growing filament barbed ends [18]. A subset of formins referred to as Diaphanous-related formins (Drfs) act as effectors of Rho family GTPases [19]. In these formins, the FH1 and FH2 domains are flanked by additional regulatory domains at the N-terminus, and by a Diaphanous-autoregulatory domain (DAD) at the C-terminus. Binding of active Rho family proteins to the GTPase-binding domain (GBD) triggers the activation of formins by disrupting the intramolecular interaction between the DAD and the N-terminal Diaphanous-inhibitory domain (DID). Several formins such as mDia2 [20,21] and the formin-like family members 2 (FMNL2) and 3 (FMNL3) [22,23,24] localize at lamellipodia and filopodia tips and have been implicated in driving these protrusions.

The second group of proteins enhancing the rate and extent of actin filament elongation are Ena/VASP proteins. Vertebrates express Vasodilator-stimulated phosphoprotein (VASP), mammalian Enabled (Mena), and Ena/VASP-like (Evl). All family members are tetramers with a tripartite architecture harboring domains allowing for interactions with FPPPP-containing receptors mediating subcellular positioning, actin monomers, profilin–actin complexes and actin filaments [10]. In contrast to formins, single Ena/VASP tetramers are only poorly processive and barely antagonize CP [25,26]. However, processivity and resistance against CP increases dramatically upon oligomerization or clustering [25,27,28]. Previously, the clustering of VASP was thought to be mediated by the membrane deforming and curvature sensing I-BAR (inverse Bar-domain) proteins such as IRSp53 (Insulin Receptor Substrate of 53 kDa) [29] or the scaffolding protein Lamellipodin (Lpd; also known as Raph1) [30]. Very recently, however, it was shown that Ena/VASP clustering involves Lpd but not I-BAR proteins and absolutely requires unconventional myosin-X (also known as Myo10 or MyoX) [31]. Ena/VASP proteins localize to sites of active actin assembly including focal adhesions (FA), SFs, the surface of bacterial pathogens and the tips of cellular protrusions [10]. Consistent with their accumulation at the tips of filopodia, genetic removal of all three Ena/VASP proteins was reported to severely perturb filopodia formation in neuronal cells on poly-L-lysine, albeit the phenotype was less drastic on laminin [32,33].

Despite being composed of actin bundles and sharing common components, microspikes represent distinct molecular entities, as revealed by more recent work [10,34,35]. Microspikes, frequently capable of surfing along the plasma membrane, are an integral part of the lamellipodium that mostly do not extend beyond the periphery of the membrane [36], whereas filopodia, capable of protruding a few microns beyond the membrane [37], can also form in the absence of lamellipodia and arise on the entire cell surface. Notably, Ena/VASP-deficient B16-F1 cells virtually lack microspikes, but they can form numerous filopodia upon treatment with the Arp2/3 complex inhibitor CK666 or ectopic expression of MyoX and active mDia2 [34]. However, in spite of their ability to induce filopodia, expression of either MyoX or active mDia2 fails to rescue microspike formation in these mutants. In line with this notion, loss of MyoX completely abolishes microspikes in B16-F1 cells [31], whereas the knockdown of MyoX in endothelial cells decreases filopodia number only by 50% [38].

The depletion of CP-ß in B16-F1 melanoma cells was previously shown to perturb lamellipodia and cause an explosive formation of filopodia [39]. Since this phenotype was very similar in Rat2 and NIH 3T3 fibroblasts, but was not observed in Ena/VASP-deficient MV^D7^ fibroblasts, it was hypothesized that Ena/VASP proteins are essential for filopodium formation in the absence of CP [39]. To this end, in this study, we employed B16-F1 and NIH 3T3 cells and CRISPR/Cas9 technology followed by comprehensive analysis of filopodium formation in derived mutant cells to experimentally test this hypothesis.

## 2. Materials and Methods

### 2.1. Constructs

Full-length *Capzb* encoding murine capping protein subunit beta (CapZβ) was amplified from an NIH 3T3 cDNA library and inserted into XhoI and EcoRI sites of plasmids pEGFP-C1 and pEGFP-N1 (Clontech, Palo Alto, CA, USA). To allow for generation of stably transfected cell lines, *Capzb* was also inserted into the XhoI and EcoRI sites of a pEGFP-C1 plasmid variant containing a Puromycin cassette (pEGFP-C1-Puro) [34]. The plasmid for the expression of LifeAct fused to EGFP has been described [34,40].

For the generation of antigens, the coding sequence encompassing residues 1–527 of murine FMNL2 was inserted into the BglII and SalI sites of pGEX 6P3 (GE Healthcare, Munich, Germany). Accordingly, the sequence encoding residues 533–1028 of murine FMNL3 was inserted into the BamHI and SalI sites of pGEX 6P1, and the sequence encoding residues 94–473 of murine mDia2 was inserted into the EcoRI and SalI sites of pGEX 6P1. All constructs were validated via sequencing.

### 2.2. Cell Culture and Transfection

NIH 3T3 fibroblasts (ATCC CRL-1658), B16-F1 mouse melanoma cells (ATCC CRL-6323) and derived mutants were cultivated at 37 °C and 5% CO_2_ in high-glucose DMEM culture medium (Lonza, Cologne, Germany) supplemented with 1% penicillin-streptomycin (Biowest, Nuaille’, France), 10% FBS (Biowest) and 2 mM UltraGlutamine (Lonza). A total of 3 h after seeding onto 35 mm diameter wells (Sarstedt, Nümbrecht, Germany), the cells were transfected with 1 µg (B16-F1) or 3 µg (NIH 3T3) plasmid DNA using JetPRIME transfection reagent (PolyPlus) at a ratio of 1 µg of DNA to 2 µL of the transfection reagent, according to the manufacturer’s protocol. At 4 h post transfection, the transfection mixture was replaced with fresh culture medium. Cells stably transfected with pEGFP-C1-Puro-CapZβ were maintained with, additionally, 1.5 µg/mL puromycin. Used cell lines were routinely authenticated following common guidelines by local authorities. 

### 2.3. Genome Editing by CRISPR/Cas9

To generate sgRNAs of 20 nucleotides with high efficiency and minimal off-target effects covering all possible splice variants, the DNA target sequence was pasted into a CRISPR/Cas9 design tool (https://cctop.cos.uni-heidelberg.de/, accessed on 23 April 2020). For *Capzb*, the targeting sequence 5′-ACTGCGCCTTGGACCTGATG-3′ was used to target exon 2 of the gene. The sequence 5′-GTGATCCGAGCTCACATCTT-3′ was used to target exon 22 of the *MYO10* gene. For the *Fmnl2* and *Fmnl3* genes, the sequences 5’-TATGGGGAGGGTTCTTCACC-3′ and 5′-TCTTGGACCCCAATGTAACA-3′, for targeting exon 3 in the respective formins, were used. Each sequence was inserted into the BbsI site of the pSpCas9(BB)-2A-Puro(PX459)V2.0 (Addgene plasmid ID: 62988) expression plasmid [41]. Validation of the target sequences was performed by sequencing with a 5′-GGACTATCATATGCTTACCG-3′ primer. At 24 h post transfection, cells were selected in cell culture medium containing 2 µg/mL (for B16-F1-derived clones) or 3.5 µg/mL (for NIH 3T3-derived clones) puromycin for 4 days and then cultivated for 24 h in the absence of puromycin. For isolation of clonal knockout cell lines, single cells were seeded by visual inspection into 96-well microtiter plates and expanded in conditioned culture medium composed of used and sterile filtered medium and fresh medium at a ratio of 1:3. Clones were verified on the genomic level using the TIDE sequence trace decomposition web tool (https://tide.nki.nl/, accessed on 26 May 2020; [42]) and on the protein level by immunoblotting using specific antibodies.

### 2.4. Protein Purification

GST-tagged mDia2 and FMNL2/3 were expressed in *E. coli* host Rosetta 2 (Sigma, St. Louis, MO, USA) by induction with 1 mM isopropyl-β-D-thiogalactoside (Carl Roth, Karlsruhe, Germany) at 24 °C for 12 h. The bacteria were harvested and lysed by ultrasonication in lysis buffer containing PBS, pH 7.4 supplemented with 2 mM DTT, 1 mM EDTA, 5 mM benzamidine (Carl Roth), 0.1 mM AEBSF (AppliChem, Darmstadt, Germany), Benzonase (1:1000, Merck, Darmstadt, Germany) and 5% (*v*/*v*) glycerol. The proteins were then purified from bacterial extracts by affinity chromatography using glutathione-conjugated agarose (Macherey-Nagel, Düren, Germany) and eluted from the column with 20 mM reduced glutathione (Carl Roth) in PBS supplemented with 2 mM DTT, 1 mM EDTA, 5 mM benzamidine, 0.1 mM AEBSF and 5% (*v*/*v*) glycerol using standard procedures. The GST tag was cleaved off by PreScission protease (GE Healthcare), and the GST tag was then absorbed on fresh glutathione-conjugated agarose. The proteins in the flow through were further purified by size-exclusion chromatography (SEC) using a HiLoad 26/600 Superdex 200 column (GE Healthcare) controlled by an Äkta Purifier System. The fractions containing mDia2, FMNL2 and −3 were pooled and dialyzed against immunization buffer (150 mM NaCl, 25 mM Tris/HCl, pH 8.0) for generation of polyclonal antibodies. 

### 2.5. Antibodies

Polyclonal antibodies against recombinant fragments of mDia2, FMNL2 and FMNL3 were raised by immunization of female New Zealand white rabbits. This was followed by antigen-affinity purification against the same antigens coupled to CNBr-activated sepharose 4B (GE Healthcare) according to the manufacturer’s protocol. Briefly, after passing the sera over the column, the resin was washed with 300 mL of PBS, pH 7.4 and bound antibodies were eluted with 0.1 M acetic acid, pH 2.3. The fractions containing the antibodies were pooled, and the pH was immediately adjusted to 7.5 using 1 M Tris/HCl, pH 8.5 followed by dialysis against PBS buffer supplemented with 55% (*v*/*v*) glycerol for long-term storage at −20 °C. The immunization of rabbits for the generation of polyclonal antibodies was conducted in accordance with national guidelines for the care and maintenance of laboratory animals and approved by the Hannover Medical School Institutional Animal Care Facility and the Lower Saxony State Office for Consumer Protection and Food Safety (LAVES) under the application number 18A255 to J.F. For immunoblotting, polyclonal antibodies against FMNL2 (1:1000 dilution), FMNL3 (1:1000, dilution) and MyoX (1:1000 dilution, [31]) and the monoclonal antibodies directed against the capping protein α1/α2 subunits mAb B5 12.3 (1:4 hybridoma supernatant, Developmental Hybridoma Bank (deposited by J. Cooper), Iowa City, IA, USA), the capping protein β2 subunit mAb 3F2.3 (1:4 hybridoma supernatant, Developmental Hybridoma Bank (deposited by J. Cooper)), pan anti-actin (1:1000, # ab119952, Abcam, Boston, MA, USA) and GAPDH (1:50,000; #CB1001-500UG, Merck, Darmstadt, Germany) were used. Primary antibodies in immunoblots were either visualized using phosphatase-coupled anti-rabbit (1:1000, #115-055-114, Dianova, Hamburg, Germany) and anti-mouse (1:1000; #115-055-62, Dianova) antibodies or by enhanced chemiluminescence using peroxidase-coupled anti-mouse IgG (Dianova; #115-035-062; 1:10,000 dilution).

For immunofluorescence, the following primary antibodies were used: rabbit anti-VASP (1:1000 dilution, [34]), rabbit anti-FMNL2 (1:1000 dilution), rabbit anti-FMNL3 (1:1000, dilution), rabbit anti-MyoX (1:1000 dilution [31]), rabbit anti-WAVE2 (1:1000 dilution [31] and the monoclonal anti-vinculin antibody hVIN-1 (1:1000 dilution; #V9131, Sigma). Primary antibodies were visualized in immunocytochemistry with Alexa-488-conjugated goat-anti-rabbit (1:1000 dilution; #A-11034, Invitrogen) or goat-anti-mouse antibodies (1:1000 dilution; #A-11029, Invitrogen). To enhance EGFP signals, Alexa488-conjugated anti-EGFP nanobodies (1:1000 dilution, [43]) were used. 

### 2.6. Immunoblotting

For the preparation of whole-cell lysates, cells were cultivated to 80–100% confluency and trypsinized. Cell pellets were washed twice with cold PBS and lysed with 400 µL of cold RIPA buffer (150 mM NaCl, 1.0% Triton-X-100, 0.5% sodium deoxycholate, 0.1% sodium dodecyl sulfate (SDS), 50 mM Tris, pH 8.0), then supplemented with 5 mM benzamidine (Carl Roth), 0.1 mM AEBSF (AppliChem, Darmstadt, Germany) and benzonase (1:1000, Merck, Darmstadt, Germany) for 25 min at 4 °C on a wheel rotator. Subsequently, the SDS concentration was adjusted to 0.3% final concentration, and cells were further lysed for 20 min at 4 °C on a wheel rotator. Then, cell lysate was centrifuged at 4 °C for 5 min at 20,000× *g*, and the supernatant was resuspended in an appropriate volume of 3 × SDS sample buffer (150 mM Tris/HCl, pH 6.8, 30% glycerol, 3% SDS, 3% β-mercaptoethanol).

Total proteins of cell lysates were resolved by SDS-PAGE, transferred by semi-dry blotting using a Trans-Blot SD Semi-Dry Transfer Cell (Bio Rad, Feldkirchen, Germany) at 16 V onto nitrocellulose membranes (Hypermol, Hannover, Germany) and blocked with NCP buffer (10 mM Tris/HCl pH 8.0, 150 mM NaCl, 0.05% Tween-20, 0.02% NaN_3_) containing 4% bovine serum albumin (BSA) for at least 30 min. Primary antibodies were incubated overnight in NCP buffer. After washing of the membranes five times with NCP buffer and incubation with secondary, phosphatase-conjugated antibodies for at least 4 h, the blots were developed with 20 mg/mL 5-brom-4-chlor-3-indolylphosphate-p-toluidin (BCIP) in 0.1 M NaHCO_3_, pH 10.0. For quantification of proteins, the immunoblots were developed by enhanced chemiluminescence, and analysis was performed with ImageJ [44]. Uncropped scans of immunoblots and gels are shown in Appendix A.

### 2.7. Immunofluorescence and Imaging

For immunofluorescence labeling, cells were fixed for 20 min in pre-warmed PBS, pH 7.3 containing 4% PFA and 0.06% picric acid and, subsequently, were washed three times with PBS supplemented with 100 mM glycine. The cells were then permeabilized with 0.1% Triton X-100 in PBS for 35 s and blocked with PBG (PBS, 0.045% cold fish gelatin and 0.5% BSA). Primary antibodies were incubated overnight, followed by washing of the specimens five times with PBG and incubation with respective secondary antibodies for at least 2 h. F-actin was visualized with Atto550-phalloidin (1:250 dilution, #AD 550–82, Atto-Tec, Siegen, Germany), and DNA was visualized with 4′,6-diamidino-2-phenylindole (DAPI) (1:1000 dilution, Sigma). The 16-bit images of fixed cells were captured with an Olympus XI-81 inverted microscope equipped with an UPlan FI 100×/1.30 NA oil immersion objective or a Zeiss LSM 980 with Airyscan 2 equipped with a Plan-Apochromat 63×/1.4 NA oil DIC objective using 405 nm, 488 nm and 594 nm laser lines. Emitted light was detected in the wavelength ranges of 410–473 nm, 490–561 nm and 576–695 nm, respectively.

Time-lapse imaging of live cells was performed using an Olympus XI-81 inverted microscope (Olympus, Hamburg, Germany) driven by Metamorph software (Molecular Devices, San Jose, CA, USA) and equipped with objectives specified below and a CoolSnap EZ camera (Photometrics, Tucson, AZ, USA). Cells were seeded onto 35 mm glass bottom dishes (Ibidi, Planegg-Martinsried, Germany), coated for 1 h with either 25 mg/mL laminin (Sigma), in the case of B16-F1 cells and derived clones, or with 20 µg/mL fibronectin (Roche, Penzberg, Germany), in the case of NIH 3T3 fibroblasts and their derivatives, maintained in imaging medium composed of F-12 Ham Nutrient Mixture with 25 mM HEPES (Sigma) to compensate for the lack of CO_2_ and supplemented with 10% FBS (Biowest), 1% Penicillin-Streptomycin (Biowest), 2 mM stable L-glutamine (Biowest) and 2.7 g/L D-glucose (Carl Roth, Karlsruhe, Germany) in an Ibidi Heating System at 37 °C. For 2D random motility assays, B16-F1 cells were seeded at low density (~5 × 10^4^ cells/mL) onto the dishes and allowed to adhere for 3 h. Subsequently, the medium was exchanged with imaging medium, the chamber was mounted into a heating system and cells were recorded by time-lapse, phase-contrast imaging with 16-bit image depth at 60 s intervals for 3 h using an UPlan FL N 4×/0.13 NA objective (Olympus) with additional 1.6× optovar magnification. NIH 3T3 and derived cells were allowed to adhere for 3 h after seeding and were imaged at 10 min intervals for 10 h using an Uplan FL N 4×/0.13 NA objective (Olympus, Hamburg, Germany). Single-cell tracking was performed with MTrackJ in ImageJ [44,45]. Analyses of cell speed and cell trajectories, turning angles and mean square displacements were performed in Excel (Microsoft, Redmond, WA, USA) using a customized macro [46]. Cells that contacted each other or divided were excluded from analysis. The directionality index was calculated by dividing the shortest distance between starting and end points (d) by the actual cell trajectories (D). Behavior of migrating cells on laminin at high magnification was recorded by time-lapse imaging at 5 s intervals for 15 min using an UPlan FI 100×/1.30 NA oil immersion objective (Olympus).

### 2.8. Quantification of F-Actin Content by Flow Cytometry

Flow cytometry was performed with an FACSAria III Fusion flow cytometer (Becton Dickinson, Franklin Lake, NJ, USA) driven by FACSDiva Vers. 8.0.1. software. For fluorescence-activated cell sorting of reconstituted CP-KO cells stably expressing EGFP-CapZβ2, EGFP was excited with a laser at 488 nm, and emission at 530 nm was detected with a band-pass filter. EGFP-positive cells were collected into a sterile 15 mL screw-cap tube containing cell culture medium with 1.5 µg/mL puromycin for further cultivation. To quantify cellular F-actin content, the cells were trypsinized, fixed and permeabilized in solution and then counted, and approximately 2 × 10^5^ cells were stained for 2 h with 3 µM Atto550-conjugated phalloidin (Attotec, Siegen, Germany) in 400 µL PBS. After washing five times with PBS, the cells were analyzed by flow cytometry using 561 nm laser light excitation. Emission at 586 nm was detected using a band-pass filter. The number of events counted was approximately 50,000 for each cell line in each run. Further analysis was performed with FlowJo software (Becton Dickinson). Data are presented as the x-fold change in phalloidin intensity in comparison to B61-F1 control.

### 2.9. Quantification of the F- to G-Actin Ratio and of Global Actin Levels

To determine the F- to G-actin ratio in B16-F1 wild-type and CP-deficient cells, nearly confluent cultures, corresponding to about 4–5 × 10^6^ cells, were first rinsed twice with ice-cold PBS. Cells from each plate were directly detached from the plate with a cell scraper using 300 µL of cold lysis buffer containing 20 mM Hepes, pH 7.2, 100 mM NaCl, 10 mM KCl, 2 mM MgCl_2_, 5 mM KPO_4_, 5 mM EGTA, 5 mM ATP, 2 mM DTT, 5% sucrose, 0.5% Triton X-100, 0.5% NP-40, 5 mM benzamidine, 0.1 mM AEBSF and 0.75 µM phalloidin (Sigma). The crude lysates were then transferred into 1.5 mL reaction tubes and placed on a rotary shaker for further homogenization at 4 °C. After 30 min, 200 µL samples were centrifuged at 150,000× *g* for 1 h using an Optima tabletop ultracentrifuge (Beckman Instruments, Palo Alto, CA, USA). Subsequently, the samples of the supernatant and the pellet fractions were brought to 300 µL with SDS sample buffer, and aliquots were subjected to SDS-PAGE followed by Coomassie Blue staining or immunoblotting. The blots were probed with pan anti-actin antibody at a 1:1000 dilution overnight. Primary actin antibodies in these immunoblots were visualized by enhanced chemiluminescence using the ChemiDoc MP Imaging System driven by Image Lab software (Biorad, Hercules, CA, USA). After background subtraction of 16-bit images, the amount of actin in the pellet and supernatant fractions was determined densitometrically using ImageJ software. For quantification of global actin levels, proteins in total cellular lysates were separated by SDS-PAGE and blotted on nitrocellulose, and blots were incubated with anti-actin and anti-GAPDH antibodies. After background subtraction of 16-bit images, band intensities of actin were normalized to respective GAPDH signals, and relative protein levels were calculated in CP-KO mutants compared with the B16-F1 control.

### 2.10. Quantification of Peripheral Filopodia

Brightest point projection of 3D reconstructions from confocal Airyscan sections were used to visualize filopodia of phalloidin-stained cells at high resolution, regardless of their morphology. Filopodium length was defined as the distance of the filopodium shaft extending beyond the periphery of the plasma membrane and its distal tip. The length and the number of peripheral filopodia per cell were quantified manually with ImageJ.

### 2.11. Analysis of Focal Adhesions

FA parameters of NIH 3T3 fibroblasts and derived mutants were inferred from confocal images of vinculin-stained cells using a customized ImageJ macro to facilitate analysis. Semi-automatic processing of 16-bit images recorded at identical settings included gradual background subtractions using a rolling ball radius of 50 pixels to obtain pre-processed images for subsequent analysis of intensity profiles. For segmentation of FAs, the resulting images were further processed by two additional background subtraction steps using rolling ball radii of 15 pixels followed by binarization of obtained images using the Otsu thresholding method [47]. The regions obtained were redirected to the pre-processed images, which were then analyzed with the analyze particles plugin in ImageJ using a minimum particle size of 0.25 µm^2^. This yielded binary masks of FAs for visual display and specific FA parameters such as number, size, length and intensity, which were further analyzed using Microsoft Excel.

### 2.12. Statistical Analyses

Quantitative experiments were performed at least in triplicates to minimize environmental bias or unintentional error. Raw data were processed in Excel. Statistical analyses were performed with Origin 2021 (OriginLab Corporation, Northampton, MA, USA). All datasets were tested for normality by the Shapiro–Wilk test. Statistical differences between normally distributed datasets of two groups were determined by *t*-test, and non-normally distributed datasets of two groups were determined by a non-parametric Mann–Whitney U rank sum test. For comparison of more than two groups, statistical significance of normally distributed data was examined by one-way ANOVA and a Tukey Multiple Comparison test. In the case of non-normally distributed data, the non-parametric Kruskal–Wallis test and Dunn’s Multiple Comparison test were used. Statistical differences were defined as * *p* ≤ 0.05, ** *p* ≤ 0.01, *** *p* ≤ 0.001 as well as n.s., not significant, and are displayed and stated in figures and figure legends, respectively.

## 3. Results

### 3.1. Loss of CP in B16-F1 Cells Triggers the Massive Formation of Filopodia and Impairs 2D-Cell Migration

To test the hypothesis of whether Ena/VASP proteins are essential for filopodia formation in CP-deficient cells [39], we first disrupted the single gene (*CapZb*) encoding the ß-subunit of CP in B16-F1 mouse melanoma cells using CRISPR/Cas9 technology. Respective protein loss in independent clonal cell lines was validated by TIDE sequence trace decomposition analyses of amplified genomic target sites (Appendix A) [42] and, finally, confirmed by sequencing of the target site and immunoblotting (Figure 1A). Given that CP operates only as a functional heterodimer, consistent with previous work [39,48], loss of the ß-subunit also resulted in a markedly reduced expression level of the α-subunit. We then analyzed actin filament (F-actin) distribution in B16-F1 and CP-KO cells after phalloidin staining. As opposed to B16-F1 control, where almost 50% of the cells displayed prominent smooth lamellipodia with numerous microspikes, approximately 12–14% of the CP-KO cells developed strongly compromised lamellipodia with bulbous, exceptionally strong ruffling cell fronts with many protruding filopodia (Figure 1B and Appendix A). The remaining 86–88% of the mutant cells were apparently not polarized and also exhibited massive formation of highly dynamic filopodia around the entire cell periphery and on the dorsal surface (Appendix A). To ensure that the observed mutant phenotypes were CP-specific, we then analyzed reconstituted CP-KO mutant cells ectopically expressing the ß2-subunit of CP fused to EGFP. Re-expression of CP-ß2 not only restored accumulation of the endogenous CP-α-subunit (Figure 1C) but also rescued cell morphology and F-actin distribution (Figure 1D).

Since 2D migration on flat surfaces is primarily driven by actin assembly in the lamellipodium, we then used phase-contrast, time-lapse imaging to measure rates of cell migration in parental B16-F1 cells and two independent CP-KO cell lines on laminin (Appendix A). As opposed to control cells, which migrated at 1.47 ± 0.38 µm/min (mean ± SD), cell speed was reduced by about 70% in both CP-deficient mutant cell lines (clones #9 and #22) to 0.40 ± 0.25 µm/min and 0.54 ± 0.28 µm/min, respectively (Figure 1E). This was accompanied by increases in directionality in both CP-KO mutants, making them 55% or 32% more directional compared to the wild type (Figure 1F). Finally, we calculated the mean square displacement (MSD) in wild-type and mutant cells to assess their effective directional movement. Despite their higher directionality, presumably owing to their markedly slower motility, both CP-deficient cell lines displayed drastically lower MSD values as compared to the B16-F1 control (Figure 1G). Taken together, these results are consistent with previous work [39,48,49,50] showing that CP is critical for effective 2D-cell migration, even though complete loss of CP, as shown here, caused more severe motility defects as compared to those previously reported for CP-depleted cells [49,50].

### 3.2. Loss of CP in B16-F1 Cells Increases Global Levels of Filamentous and Total Actin

In agreement with previous work analyzing CP-depleted *Dictyostelium* or B16-F10 mouse melanoma cells [49,50], elimination of CP in B16-F1 cells caused a significant increase in F-actin intensity, as evidenced by epifluorescence imaging of phalloidin-stained wild-type and mutant cells at identical settings (Figure 2A). To allow the most accurate quantification of F-actin content in CP-deficient B16-F1 cells, we trypsinized adherent wild-type and mutant cells and fixed and stained them with fluorescent phalloidin at saturating conditions in solution. The labeled specimens were then analyzed by flow cytometry, which is a powerful tool for the precise quantitation of fluorescence intensities on the single-cell level [51]. As expected, both CP-KO mutants exhibited markedly increased phalloidin fluorescence intensities as compared to the B16-F1 control (Figure 2B). Quantification of relative fluorescence intensities revealed that the F-actin content was increased 4- to 5-fold in the CP-KO cells compared with the B16-F1 control (Figure 2C), supporting the notion that elimination of CP elicits unleashed actin assembly. Notably, re-expression of EGFP-labeled CP-ß in CP-KO cells almost fully reverted phalloidin intensity in reconstituted cells to wild-type levels (Figure 2D). Given the drastically increased F-actin levels in the mutants, we then asked how loss of CP would affect the F- to G-actin ratio in the CP-KO cells. The amounts of free globular G-actin and filamentous F-actin were determined by differential precipitation of F-actin (pellet fraction) and G-actin (supernatant fraction) in total cell lysates after ultracentrifugation. After SDS-PAGE and Coomassie Blue staining, we noticed a prominent band at 42 kDa in the pellet fractions of the mutant cells, presumably representing actin (Figure 2E). This was confirmed by immunoblotting with an anti-actin antibody. Densitometric analysis and quantification of respective actin signals revealed a dramatically increased F- to G-actin ratio of more than 7:1 in both CP-KO mutants, as opposed to a ratio of 1.2:1 in the B16-F1 control (Figure 2E). Given the critical function of CP in maintaining the available pool of polymerization-competent monomeric actin, finally, we analyzed, by quantitative immunoblotting, whether the markedly increased F-actin content in CP-KO cells also altered global actin expression. Densitometry revealed that total actin levels, normalized to GAPDH expression, were approximately 2.8-fold higher in the KO mutants compared to the B16-F1 control (Appendix A). Taken together, these data suggest that, due to excessive actin polymerization, CP-deficient cells apparently compensate for depletion of the G-actin pool by stronger expression of actin.

### 3.3. Loss of CP in NIH 3T3 Fibroblasts Induces the Massive Formation FAs and SFs Instead of Filopodia on Fibronectin

To corroborate the filopodial and migratory phenotypes after loss of CP in a more strongly adherent cell type, we then disrupted the ß-subunit of CP by CRISPR/Cas9 in NIH 3T3 fibroblasts. Respective loss of CP-ß in independent clonal cell lines was again validated by TIDE analysis of amplified genomic target sites and, finally, confirmed by immunoblotting (Figure 3A and Appendix A). Comparable to CP-deficient B16-F1 cells, in NIH 3T3 cells, loss of the ß-subunit of CP also resulted in markedly reduced expression or stability of the α-subunit (Figure 3A). We then examined F-actin distribution in parental NIH 3T3 cells and derived CP-KO cells after phalloidin staining. However, in variance to previous work [39], and as opposed to CP-deficient B16-F1 cells, CP-deficient NIH 3T3 cells did not form numerous filopodia but instead developed a conspicuously dense meshwork of SFs that almost completely filled the mutant cells (Figure 3B), suggesting that CP-KO cells become extraordinarily adhesive [52]. To experimentally test this hypothesis and examine the consequences on fibroblast adhesion after loss of CP, wild-type and CP-deficient NIH 3T3 cells migrating on fibronectin were labeled for the focal adhesion (FA) marker protein vinculin and assessed for various features (Figure 3C). To this end, images captured at identical settings were processed into binary images using a customized macro, allowing global and unbiased assessment of multiple FA parameters. Notably, vinculin intensity was markedly increased in the CP-deficient mutants by 54.1 ± 36.1% (clone #4) and 42.2 ± 33.4% (clone #9), as compared to the parental NIH 3T3 cell line (Figure 3D). Moreover, the number of FAs in CP-KO cells was also increased by more than 50%, to 302 ± 128 and 287 ± 104, as compared to the wild type, with 138 ± 43 (Figure 3E). Quantification of FA size, furthermore, revealed an increase in CP-KO cells by more than 50%, to 1.7 ± 0.4 µm^2^ and 1.7 ± 0.4 µm^2^, as compared to NIH 3T3 control cells, with 1.1 ± 0.3 µm^2^ (Figure 3F). In addition, FAs in the mutant cells were considerably longer and wider as compared to control (Appendix A). Thus, loss of CP in NIH 3T3 fibroblasts apparently appears to dramatically improve adhesion to their preferred cell substrate fibronectin but does not lead to excessive filopodia formation.

Finally, we assessed the consequences of CP deficiency on fibroblast 2D-cell migration on fibronectin by phase-contrast, time-lapse imaging of NIH 3T3 and CP-KO cells (Figure 3G and Appendix A). In both CP-KO mutants, cell speed (0.10 ± 0.06 µm/min and 0.11 ± 0.06 µm/min) was significantly reduced as compared to NIH 3T3 control (0.29 ± 0.08 µm/min). Consistently, the CP-KO mutants exhibited much lower MSD values when compared to the NIH 3T3 wild type (Figure 3H). Thus, despite their highly different phenotypes regarding filopodia formation, both B16-F1- and NIH 3T3-derived mutants exhibited strongly compromised 2D-cell migration after loss of CP. 

### 3.4. CP-Deficent NIH 3T3 Cells form Numerous Filopodia on Poorly Adhesive Substrates

The strongly amplified formation of integrin-based cell adhesions, as well as the excessive spreading of NIH 3T3-derived CP-KO mutants on fibronectin, suggested enhanced contractility of FA-anchored SFs, likely resulting in cells with highly stretched membranes. Interestingly, it has recently been shown that actin-based protrusions such as lamellipodia adapt to changes in membrane tension imposed by branched actin networks, such that large lamellipodia are formed at low tension, whereas their formation is perturbed at high tension [53]. Notably, we found that lamellipodia formation in NIH 3T3 fibroblasts was suppressed after loss of CP, as evidenced by immunostaining the cells with the WRC component WAVE2 (Appendix A). Thus, in addition to the relevance of CP for lamellipodium formation, this could also mean that filopodia formation in NIH 3T3-derived CP-KO cells on fibronectin is suppressed because the force generated by nascent filopodia is not sufficient to overcome the high tension of the membrane. Since it is not possible to distinguish between these two possibilities without elaborate biophysical methods such as atomic force microscopy (AFM) or the use of novel tension sensors probes reporting membrane tension changes through their fluorescence lifetime [54], we therefore asked whether filopodium formation could be induced by seeding the cells on poorly adhesive substrates. As shown in Figure 4, CP-KO cells seeded on poly-L-lysine or directly on glass adhered poorly but formed numerous filopodia, comparable to CP-deficient B16-F1 cells plated on laminin. Thus, the striking filopodia phenotype upon loss of CP appears to be cell-type- and context-dependent.

### 3.5. Loss of CP in Ena/VASP-Deficient B16-F1 Cells Does Not Abrogate Filopodium Formation

In previous work, it has been hypothesized that Ena/VASP proteins are obligatory for filopodium formation in the absence of CP [39]. To confirm or challenge this hypothesis, we first examined VASP localization in CP-deficient B16-F1 cells. CP-KO cells stained for the F-actin cytoskeleton and endogenous VASP indeed showed conspicuous VASP clusters at the distal tips of filopodia (Figure 5A). Then, we disrupted the ß-subunit of CP by CRISPR/Cas9 in Ena/VASP/Mena-deficient B16-F1 cells (EVM-KO), recently shown to virtually lack microspikes but still be capable of forming filopodia upon transient expression of active mDia2 and unconventional myosin-X or by pharmacological inhibition of the Arp2/3 complex [34]. Loss of CP-ß in independent clonal cell lines, denoted as EVM/CP-KO cells, was again validated by TIDE analysis of amplified genomic target sites and confirmed by immunoblotting (Figure 5B and Appendix A). Both independent EVM/CP-KO mutant cells (clones #4 and #14) were highly similar, but, to our surprise, they still formed numerous filopodia, similar to the CP-KO cells, as determined by confocal Airyscan imaging of cells stained with phalloidin migrating on laminin (Figure 5C and Appendix A). Quantification further revealed filopodia to be about 20% shorter in EVM/CP-KO mutants (2.2 ± 0.5 and 2.2 ± 0.5 μm) as compared to CP-KO control cells (2.8 ± 0.8 and 2.8 ± 0.7 μm) (Figure 5D). Notably, the quantification of the number of peripheral filopodia, on the other hand, showed no noticeable differences between these cell lines (Figure 5E), suggesting that Ena/VASP proteins are not causally implicated in the nucleation of filopodial actin filaments. Nevertheless, these data clearly show that even though Ena/VASP proteins contribute to filopodia formation in B16-F1 cells lacking CP, they are clearly dispensable for the generation of these structures.

### 3.6. Endogenous MyoX, FMNL2 and -3, but Not mDia2, Reside at the Tips of Filopodia in CP-KO Cells

These results clearly showed that in the absence of CP and Ena/VASP, additional factors must contribute to filopodium formation in the mutants. According to our current state of knowledge, besides unconventional MyoX [34,55,56,57], several formins, in particular, mDia2 [20,21,34], and the formin-like family members 2 (FMNL2) and 3 [22,24,58,59] were reported to localize at filopodia tips and have been implicated in driving these protrusions. We therefore asked which of these potent filopodia inducers localize at the tips of filopodia in EVM/CP-KO mutants. To this end, we used affinity-purified polyclonal antibodies and examined localization of these factors by indirect immunofluorescence in EVM/CP-KO cells. Strikingly, contrary to our expectation, endogenous mDia2 was not detected at filopodia tips in the mutant cells (Figure 6A). To exclude the possibility that the antibody is not capable of detecting mDia2 at filopodia tips by immunofluorescence, we ectopically expressed a constitutive variant of mDia2 lacking its autoinhibitory domain (mDia2ΔDAD) fused to EGFP, followed by immunolabeling with the mDia2 polyclonal antibody. Consistent with previous work [20,21], expression of active mDia2 triggered the formation of numerous filopodia. Most importantly, even in transfected cells expressing very low levels of the fusion protein, as assessed by EGFP fluorescence, active mDia2 was robustly detected at filopodia tips by the polyclonal antibody (Figure 6B), confirming its usefulness and specificity. Since ectopically expressed active mDia2 fused to EGFP was previously also reported to localize to the cell front and the tips of microspikes [20,21], we also examined localization of endogenous mDia2 in B16-F1 cells forming prominent lamellipodia. However, once again, endogenous mDia2 was neither detectable at the periphery of the leading edge nor at microspike tips (Figure 6C). Notwithstanding this, and despite the lack of mDia2 localization in B16-F1 cells during the interphase, endogenous mDia2 localized prominently together with tubulin at the mitotic spindle during prometaphase and the midbody during telophase (Figure 6D). These findings were corroborated in human HeLa cells and NIH 3T3 fibroblasts (Appendix A), supporting the view that mDia2 is rather implicated in cell division and not in the formation of filopodia. On the other hand, endogenous MyoX as well as FMNL2 and -3 were found to accumulate markedly at the tips of filopodia in EVM/CP-KO cells, with FMNL2 and -3 also detectable in dispersed clusters along the shafts of filopodial actin bundles (Figure 7).

### 3.7. Loss of MyoX in EVM/CP-KO Cells Leads to an Additional Reduction of Filopodia Length

To assess the contribution of MyoX in filopodium formation in the absence of CP and Ena/VASP, we disrupted the *Myo10* gene encoding MyoX in EVM/CP-KO cells using CRISPR/Cas9-mediated genome editing. Successful disruption of the gene in independent mutants, referred to as EVM/CP/MyoX-KO, was validated by TIDE analysis and verified by immunoblotting (Figure 8A and Appendix A). Both independent EVM/CP/MyoX-KO mutants still formed numerous filopodia in a comparable fashion, as evidenced by confocal Airyscan imaging of phalloidin-stained cells migrating on laminin, but, notably, their filopodia appeared to be considerably shorter as compared to those formed by EVM/CP-KO cells (Figure 8B). This was substantiated by quantification of filopodium length, showing filopodia of EVM/CP/MyoX-KO cells to be almost 30% shorter (1.6 ± 0.3 and 1.6 ± 0.3 µm) as compared to EVM/CP-KO control cells (2.2 ± 0.5 μm) (Figure 8C). However, as with the elimination of CP in EVM-KO cells, quantification of the number of peripheral filopodia showed no significant differences between parental EVM/CP-KO and derived EVM/CP/MyoX-KO cells (Figure 8D). Since filopodia formation may be linked to stabilization by adhesions formed with the substrate [60], we also examined whether the EVM/CP/MyoX-KO cells exhibit any changes in their spreading behavior on laminin as compared to the EVM/CP-KO cells. As shown in Appendix A, additional loss of MyoX in EVM/CP-deficient cells only marginally affected spreading of the EVM/CP/MyoX-KO mutant cells as compared to the parental cell line, strongly suggesting that the observed reduction of filopodia length is not correlated with diminished adhesion, but with the lack of MyoX activity driving filopodium formation.

### 3.8. Combined Inactivation of Ena/VASP, MyoX and FMNL2 and -3 Is Required to Drastically Impair Filopodium Formation in CP-Deficient Cells

To assess the contribution of FMNL-family formins to filopodium formation, we finally disrupted the *FMNL2* and *FMNL3* genes in EVM/CP/MyoX-KO cells using CRISPR/Cas9 technology. We initially isolated independent clonal cell lines with disrupted expression of FMNL3 alone (EVM/CP/MyoX/FMNL3-KO) followed by additional disruption of FMNL2 (EVM/CP/MyoX/FMNL2/3-KO). In total, two independently generated double-KO clones of each genotype, as validated by TIDE analysis and immunoblotting (Figure 9A and Appendix A), were further analyzed. In contrast to the EVM/CP/MyoX-KO reference, filopodia formation was severely impaired phenotypically in all FMNL-deficient mutants, with cells forming only stub-like projections that barely extended beyond the cell periphery, as revealed by imaging of phalloidin-stained cells (Figure 9B). Consistently, quantification revealed filopodia to be 1.1 ± 0.2 μm, more than 30% shorter in all FMNL-deficient mutants as compared to EVM/CP/MyoX-KO control cells, at 1.6 ± 0.30 μm (Figure 9C). Notably, analyses of phalloidin-stained cells further showed all FMNL-KO mutant cells forming roughly 30% less filopodia compared to control (Figure 9D). Thus, the removal of FMNL-family formins reduced both the length and the number of filopodia, which is consistent with their ability to nucleate and actively elongate actin filaments in vitro [22]. Unexpectedly, combined loss of FMNL3 and FMNL2 did not cause a noticeably stronger phenotype regarding filopodium formation than loss of FMNL3. Nevertheless, this is consistent with previously published work analyzing protrusion and cell migration in B16-F1 cells showing that loss of FMNL2 alone generally caused weaker phenotypes as compared to loss of FMNL3 [23].

## 4. Discussion

In this work, we examined the consequences of CP loss in mouse B16-F1 melanoma cells and NIH 3T3 fibroblasts. Consistent with previous work using RNA interference or genetic knockout in various cell types [39,48,49,50], disruption of CP-ß expression in B16-F1 cells was associated with markedly reduced levels of the CP-α-subunit and resulted in unleashed actin assembly accompanied with drastic changes in cell morphology and organization of the actin cytoskeleton. Notably, quantification of the total F-actin content determined by flow cytometry of phalloidin-stained cells revealed a 4- to 5-fold higher level in B16-F1-derived CP-KO mutants as compared to B16-F1 control. Even though increased F-actin levels have been also detected in other cell types upon perturbation of CP function [49,50], this value is significantly higher than the only 2-fold higher value previously reported for CP-depleted B16-F10 cells [49]. This suggests that the difference is presumably mainly due to incomplete removal of CP by siRNA interference, albeit it may also be affected by the different quantification methods. Given the typical ratio of F-actin to G-actin of approximately 1:1 in non-muscle cells [50,61,62], our data revealing a ratio of 7:1 in CP-KO cells further suggested a significant depletion of the polymerization-competent pool of monomeric actin in the mutants. Interestingly, we additionally found increased global actin levels in independent CP-KO cells that were almost 3-fold higher as compared with the B16-F1 wild type, suggesting that loss of CP elicits compensatory mechanisms to account for diminished G-actin levels by amplified expression of actin, for instance, by increased transcription or stability of the respective mRNA. Since actin polymerization and disassembly of actin filament plays a critical role in protrusion and cell migration [63,64], it was not surprising to find that CP-deficient NIH 3T3 and B16-F1 mutants exhibited very strong migration defects. These results are consistent with previous work analyzing the motility of CP-depleted B16-F10 and *Dictyostelium* cells [49,50], but, again, the motility defects were more severe upon complete loss of CP. These results are furthermore in line with a very recent study showing that the rate of lamellipodium protrusion is considerably impaired in CP-deficient B16-F1 cells [48].

The depletion of CP in different cell types such as B16-F1, B16-F10 and *Dictyostelium* cells is commonly linked to excessive formation of filopodia [39,49,50], as shown here for CP-deficient B16-F1 cells. However, in contrast to a previous study proposing that CP-depleted NIH 3T3 fibroblasts also form numerous filopodia [39], we found that loss of CP in independent clonal cell lines derived from NIH 3T3 fibroblasts was accompanied by excessive formation of long ventral SFs linked to exaggerated FAs, occupying the entire cell area, instead of the formation of numerous filopodia. This raises the important question of as to why CP-deficient fibroblasts form an excess of adhesive and contractile structures at the expense of filopodia. Of note, it has been recently shown that different phenotypes, referred to as “clonal variability” in genome-edited cells, are already partially attributable to the heterogeneity of wild-type cells [65]. Intriguingly, this study found hundreds of differentially regulated transcripts when comparing clonal populations of mIMCD-3 wild-type cells. Since these differences are certainly even more pronounced in diverse cell types such as the B16-F1 melanoma cells and NIH 3T3 fibroblasts used here, the specific phenotype that emerges after the loss of CP in one cell type cannot simply be transferred to the other. Interestingly, ectopically expressed CP fused to GFP has been found to localize along SFs in *Xenopus* XTC fibroblasts [66]. Since CP binds to the barbed end with subnanomolar affinity [67], thus preventing the addition and loss of actin monomers at the end, it is not surprising in our view that the loss of CP in NIH 3T3 fibroblasts promotes amplified growth of SF filaments to generate excess of exceptionally long SFs connected to enlarged FAs. Given the strongly amplified formation of FAs and SFs and the excessive spreading of NIH 3T3-derived CP-KO mutants on fibronectin, we assume that enhanced contractility of FA-anchored SFs likely leads to cells with a highly stretched membrane that has been shown to suppress the formation of protrusive structures [53]. Although this needs to be confirmed by direct measurements using AFM or membrane tether pulling, our assumption is supported by the analysis of primary adult fibroblasts derived from CP-ß-deficient mice showing that inactivation of CP leads to increased contractility and cell tension on stiff hydrogels [68]. This suggests that filopodia formation in NIH 3T3-derived CP-KO cells on fibronectin is suppressed because the pushing force of emerging filopodia is insufficient to overcome the high tension of the membrane. Consistent with this notion, we were able to induce the formation of numerous filopodia by seeding these CP-KO mutants on poorly adhesive substrates.

The most important aspect of this study was the experimental reassessment of the previous hypothesis proposing that Ena/VASP proteins are essential for filopodium formation in the absence of CP [39]. Even though we could confirm the explosive formation of filopodia in CP-deficient B16-F1 mutants, we still found comparable numbers of filopodia that were only 20% shorter in independent mutants lacking CP and all three Ena/VASP proteins (Table 1). Thus, despite providing supportive evidence for the contribution of Ena/VASP proteins to filopodia formation in this cell type, these results clearly challenge the previous hypothesis by showing that additional factors must be at play that drive the formation of these finger-shaped structures in the absence of CP and Ena/VASP. In a systematic search for these factors, we identified endogenous MyoX and the FMNL-family formins FMNL2 and FMNL 3 to reside prominently at the tips of distal filopodia in EVM/CP-KO cells. Consecutive elimination of these factors resulted in a gradual further decrease of filopodia length by 50% (Table 1), clearly demonstrating that, in addition to Ena/VASP proteins, endogenous MyoX and FMNL3 also effectively contribute to filopodia formation in the absence of CP. On the other hand, additional loss of FMNL2 had virtually no effect. Given the prominent localization of endogenous FMNL2 at the tips of filopodia, this result was rather unexpected. Nevertheless, previous work analyzing lamellipodium dynamics and cell migration of B16-F1-derived mutant cells revealed that phenotypes associated with the loss of FMNL2 are generally weaker than the loss of FMNL3 [23], which in turn may be related to the stronger nucleation activity of FMNL3 compared to FMNL2 [23]. In line with this notion, we found filopodia numbers to be decreased only in the multiple-KO mutants lacking FMNL3 and FMNL2/3 supporting the view that filopodia can arise independently of lamellipodial networks by formin-mediated de novo nucleation and elongation of filopodial filaments [37,69,70]. 

The immunolabeling of CP-deficient B16-F1 cells further revealed that endogenous MyoX, FMNL2 and 3 were like VASP already present at the tips of filopodia in CP-KO cells (Figure 5A and Appendix A), strongly supporting the view that all these factors are part of a common machinery promoting filopodium formation in CP-deficient cells. Moreover, cells have also frequently the remarkable ability to compensate for the loss of one protein by up-regulating or down-regulating others. Thus, we compared global levels of MyoX and the FMNL2/3 formins in B16-F1 cells and a set of mutants. However, we did not detect any significant changes in the global levels of MyoX or the FMNL2/3 formins in respective mutant cell lines compared to B16-F1 control (Appendix A). Combined, these findings therefore not only exclude compensatory mechanisms leading to upregulation of specific tip factors upon loss of others, but also comprehensibly explain the gradual decrease in filopodia formation upon successive elimination of individual components.

Even though ectopic expression of constitutively-activated variants of mDia2 fused to EGFP is well established to faithfully induce numerous filopodia-like structures, with the formin accumulating at their tips [21,34,71], to our surprise, we could not detect endogenous mDia2 at the tips of filopodia in EVM/CP-KO cells. Moreover, mDia2 could also not be localized at other protrusive structures such as the tips of lamellipodia or microspikes in B16-F1 cells. Given the specificity and sensitivity of the antibody, capable of detecting even minute amounts of ectopically expressed mDia2, the diffuse staining pattern rather suggests that mDia2 is largely autoinhibited during interphase, as release from autoinhibition has previously been shown to be critical for subcellular localization of various Drfs [20,72,73,74]. In marked contrast, we found that endogenous mDia2 accumulates prominently along with tubulin at the mitotic spindle and the midbody in dividing cells of different types, supporting its critical function in cytokinesis, as already suggested by previous work [75,76,77]. These studies have shown endogenous mDia2 to be associated with the midbody and the equatorial cortex, albeit the latter localization was not seen in our experiments. The reason for this partly different localization pattern is currently unclear, but could be due to the different fixation methods used. The function of mDia2 in cytokinesis is further aided by an abnormal configuration of the spindle in mDia2-depleted cells [78] as well as the analysis of mice with global knockout of mDia2, which proved to be embryonically lethal [77]. Thus, although we cannot formally exclude the presence of minute amounts of mDia2 on filopodia tips that are below the detection limit of antibody staining, collectively our data do not support a contribution of this formin to filopodia formation.

Of note, not even the combined elimination of CP, Ena/VASP proteins, MyoX, and FMNL-family formins was sufficient to completely abolish filopodia, raising the question about the remaining factors promoting the generation of the short, stub-like protrusions in the multiple KO mutants. We envision three possible scenarios to explain these results. Firstly, residual filopodia formation could be facilitated by other, yet specific tip factors that have not been analyzed in this study. This group could include other unconventional barbed-end directed myosins such as Myosin IIIa, VIIa and XVa [79,80,81,82] as well as other formins. One potential formin that has been implicated in filopodia formation is mDia1, although conflicting results have been reported in different studies. Whereas expression of constitutively active mDia1 for instance was found to induce filopodia-like protrusions in NIH 3T3 cells [74], another study using Jurkat T lymphocytes, 300.19 pre-B lymphomas and NIH 3T3 cells reached the opposite conclusion [24]. The overexpression mDia3 has been linked to IRSp53-driven filopodium formation in N1E115 cells [83], even though endogenous mDia3 could not be detected at filopodial tips. Notably, we did not detect endogenous mDia1 and mDia3 at the distal tips of filopodia in EVM/CP-KO cells with our specific antibodies (data not shown), suggesting that these formins do not contribute to filopodia formation, at least in this cell type. This notion is further supported by recent work showing that mDia1 and mDia3 are primarily involved in establishment and maintenance of the actin-rich cell cortex in the rear [84]. The last member of the formin family currently thought to contribute to filopodia formation is Daam1 [85,86]. However, in variance to other formins that localize specifically at the tips, Daam1 is an actin-bundling protein that localizes along filopodial shafts [85]. Secondly, it is also conceivable that other elongation factors, e.g., formins not normally involved in filopodia formation, occupy free barbed ends after the loss of specific filopodial factors to promote filament growth and thus give rise to filopodia. Notably, filopodia can also form by convergent elongation and coalescence of lamellipodia filaments [87]. Thus, as the third possibility we should also consider Arp2/3 complex-meditated actin assembly and distributive actin polymerase activity of the WRC, shown to accelerate elongation of uncapped actin filaments [88], which could be followed by bundling of the filaments by cross-linkers such as fascin, espin [89] or Daam1 [85], leading to short and stiff bundles capable of protruding beyond the cell periphery. Thus, the contribution all these factors in filopodium formation in the absence of CP will only be revealed by additional combinatorial loss of function studies and phenotypic side-by-side comparisons in future work.

## Figures and Tables

**Figure 1 cells-12-00890-f001:**
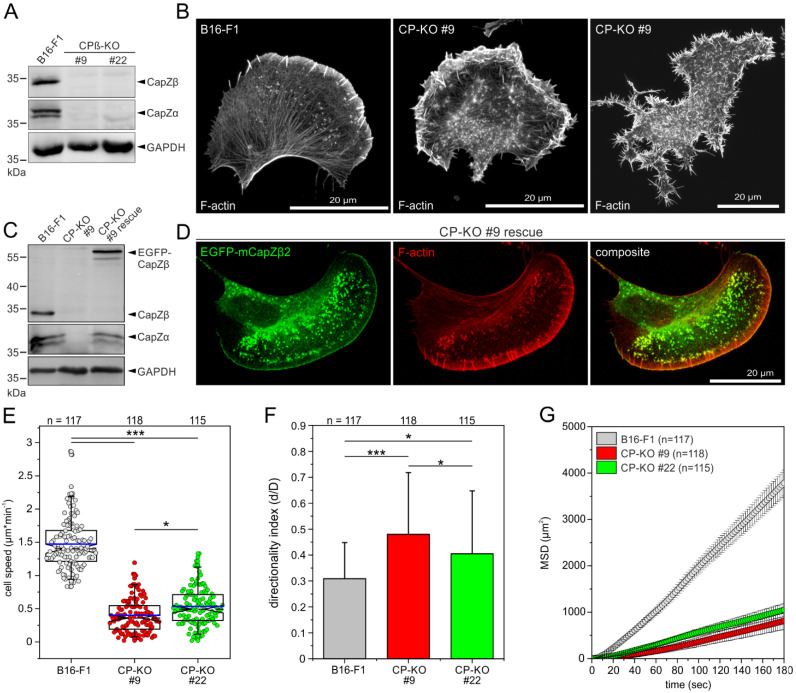
Loss of CP in B16-F1 cells triggers the massive formation of filopodia and impairs 2D-cell migration. (**A**) Immunoblot confirming the elimination of CP-ß in two independent single-knockout B16-F1 mutants (clones #9 and #22). The loss of the ß-subunit also led to the almost complete loss of the α-subunit. Loading control: GAPDH. (**B**) Morphology of representative B16-F1 cells with a smooth leading edge and embedded microspikes (left) and of CP-KO mutants that were either polarized and exhibited strong ruffling of the leading edge (middle) or were unpolarized (right). Cells migrating on laminin were stained for F-actin with phalloidin. Note the dramatically increased formation of filopodia in the mutant cells. (**C**,**D**) Reconstitution of CP-KO cells with EGFP-tagged CP-ß restores expression of the CP-α-subunit (**C**) and rescues the phenotype (**D**). (**E**) Loss of CP in B16-F1 cells decreases random 2D-cell migration on laminin. At least three time-lapse movies from three independent experiments were analyzed for each cell line. The boxes in box plots indicate 50% (25–75%) and whiskers (5–95%) of all measurements, with dashed black lines depicting the medians and arithmetic means highlighted in blue. (**F**) Directionality increased upon inactivation of CP. Bars represent arithmetic means ± SD. (**G**) Analyses of mean square displacement of wild-type versus mutant cells. Error bars represent means ± SEM. (**E**,**F**) Non-parametric, Kruskal–Wallis test and Dunn’s Multiple Comparison test were used to reveal statistically significant differences between datasets. * *p* ≤ 0.05, *** *p* ≤ 0.001; n.s.: not significant. n: number of cells analyzed from at least three independent experiments.

**Figure 2 cells-12-00890-f002:**
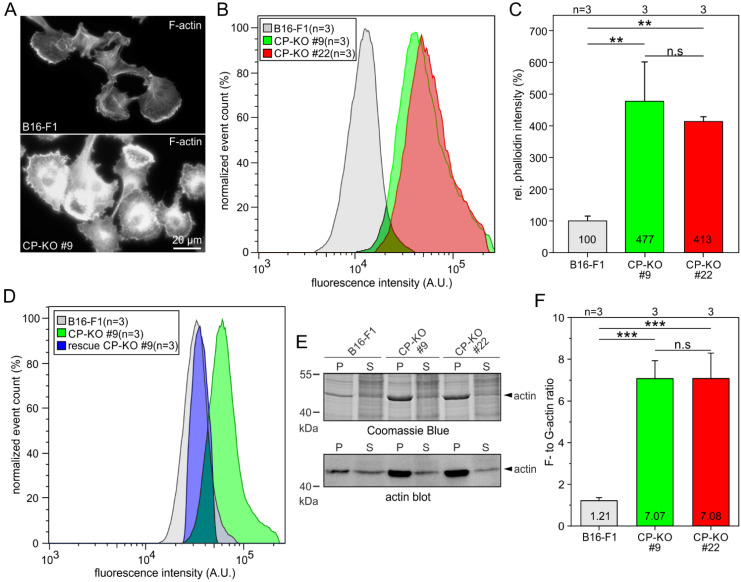
Loss of CP results in markedly increased F-actin levels. (**A**) Representative examples of B16-F1 cells and derived CP-KO mutant stained with phalloidin for the F-actin cytoskeleton and imaged at identical settings. Note the much brighter phalloidin signal in CP-KO cells. (**B**) Flow cytometry of phalloidin-stained B16-F1 cells and two independent CP-KO mutant cell lines. (**C**) Quantification of F-actin content from flow cytometry experiments shown in (**B**). (**D**) Flow cytometry of phalloidin-stained B16-F1 wild-type, CP-KO and reconstituted CP-KO cells expressing EGFP-CP-β. (**E**) Immunoblot depicting actin levels in pellet (P) and supernatant (S) fractions of B16-F1 and derived CP-KO mutants. The corresponding Coomassie Blue-stained gel is shown above. Note the prominent band of approximately 42 kDa in the pellet fractions of the mutant cells, which most likely represents actin. (**F**) Quantification of actin in pellet (P) and supernatant (S) fractions from immunoblots shown in (**E**). (**C**,**F**) Bars represent arithmetic means ± SD. Non-parametric, Kruskal–Wallis test and Dunn’s Multiple Comparison test (**C**) and one-way ANOVA and Tukey Multiple Comparison test (**F**) were used to reveal statistically significant differences between datasets. ** *p* ≤ 0.01, *** *p* ≤ 0.001; n.s.: not significant. n: number of independent experiments using approximately 5 × 10^4^ cells for each cell line (**B**,**D**) or the number of independent experiments (**C**,**F**).

**Figure 3 cells-12-00890-f003:**
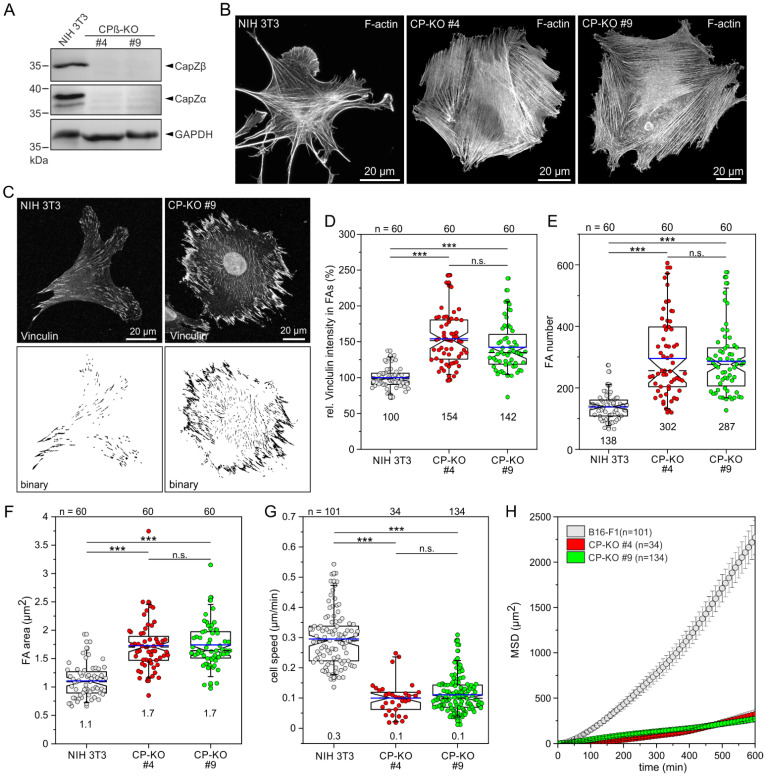
Loss of CP in NIH 3T3 fibroblasts amplifies SF and FA formation and impairs 2D-cell migration. (**A**) Immunoblot confirming the elimination of CPß in two independent single-knockout NIH 3T3 mutants. The loss of the CPß-subunit also led to the almost complete loss of the CPα-subunit. Loading control: GAPDH. (**B**) Morphologies of representative NIH 3T3 and CP-KO mutant cells migrating on fibronectin and stained for F-actin with phalloidin. Note the dramatically increased formation of SFs in the mutant cells. (**C**) Representative micrographs of NIH 3T3 and a derived CP-KO mutant cell displaying vinculin staining before (upper panel) and after processing with a customized Fiji macro (lower panel). (**D**) Quantification of vinculin intensities in FAs. (**E**) Quantification of FA number. (**F**) Quantification of FA area. (**G**) Loss of CP results in decreased cell speed. (**H**) Analyses of mean square displacement of wild-type versus mutant cells. Respective symbols and error bars represent means ± SEM. (**D**–**G**) The boxes in box plots indicate 50% (25–75%) and whiskers (5–95%) of all measurements, with dashed black lines depicting the medians and arithmetic means highlighted in blue. Non-parametric Kruskal–Wallis test and Dunn’s Multiple Comparison test were used to reveal statistically significant differences between datasets. *** *p* ≤ 0.001; n.s.: not significant. n: number of cells analyzed from at least three independent experiments.

**Figure 4 cells-12-00890-f004:**
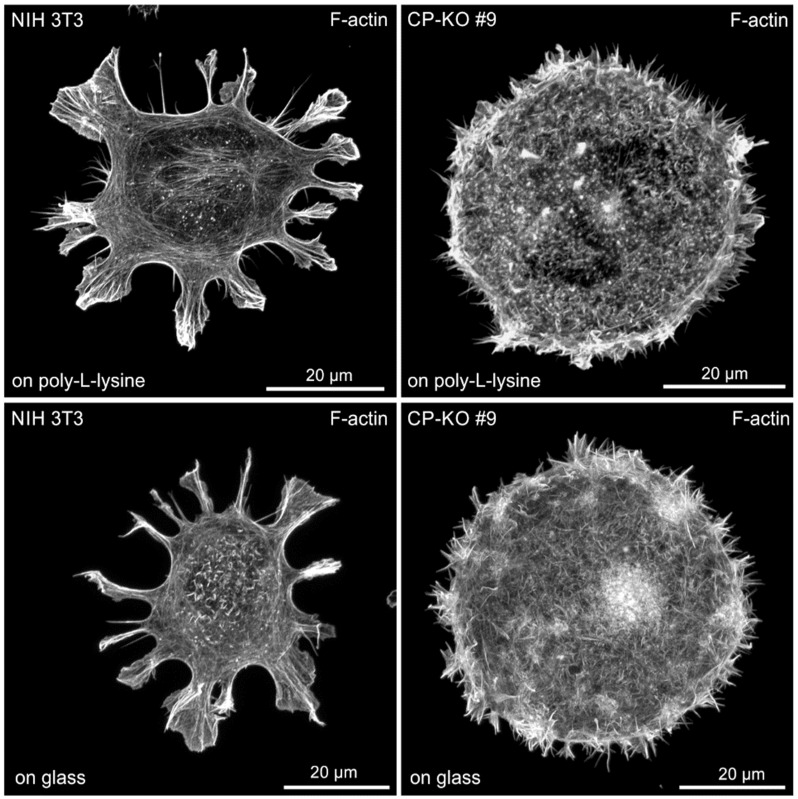
CP-deficient NIH 3T3 fibroblasts form numerous filopodia on poorly adhesive substrates. Representative images of NIH 3T3 wild-type and derived CP-KO mutant cells seeded on poly-L-lysine-coated glass (**top**) or directly on glass (**bottom**) and stained for the F-actin cytoskeleton with fluorescent phalloidin. Brightest point projection of 3D reconstructions from Airyscan confocal sections are shown.

**Figure 5 cells-12-00890-f005:**
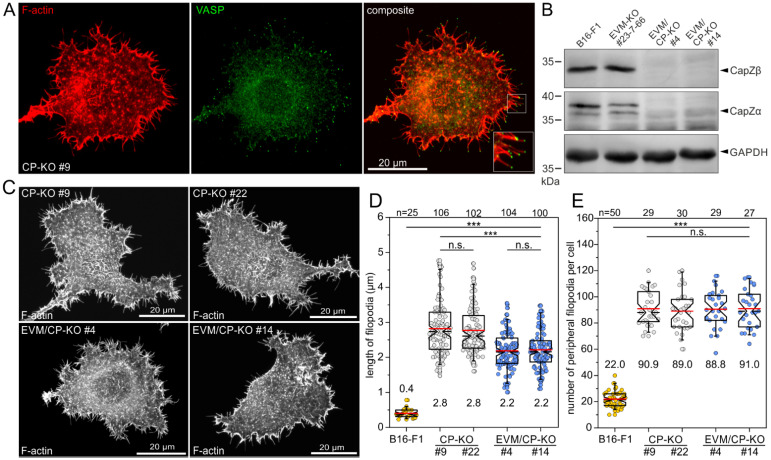
B16-F1 mutants continue to display massive filopodia formation even after combined loss of all Ena/VASP proteins and CP. (**A**) A CP-deficient B16-F1 cell stained for endogenous VASP and F-actin confirms localization of VASP at the distal tips of filopodia. Inset, enlarged image of boxed region. (**B**) Immunoblot confirming the elimination of CP-ß in two independent EVM-KO mutants (clones #4 and #14). Loss of the ß-subunit, in turn, led, again, to the almost complete loss of the CP-α-subunit. Loading control: GAPDH. (**C**) Morphologies of representative CP-KO and EVM/CP-KO mutant cells migrating on laminin and stained for the F-actin cytoskeleton with phalloidin. Note the continued formation of numerous filopodia in EVM/CP-KO cells. (**D**) Quantification of filopodia length in B16-F1 wild-type and derived CP-KO and EVM/CP-KO mutant cells. Note that B16-F1 cells, as shown in Figure 1B, primarily form microspikes and barely any filopodia that protrude beyond the periphery of the membrane. (**E**) Quantification of the number of peripheral filopodia in CP-KO and EVM/CP-KO mutant cells. (**D**,**E**) The boxes in box plots indicate 50% (25–75%) and whiskers (5–95%) of all measurements, with dashed black lines depicting the medians and arithmetic means highlighted in red. (**D**) Non-parametric Kruskal–Wallis test and Dunn’s Multiple Comparison test and (**E**) one-way ANOVA and Tukey Multiple Comparison test were used to reveal statistically significant differences between datasets. *** *p* ≤ 0.001; n.s.: not significant. n: number of filopodia (**D**) or cells (**E**) analyzed from three independent experiments.

**Figure 6 cells-12-00890-f006:**
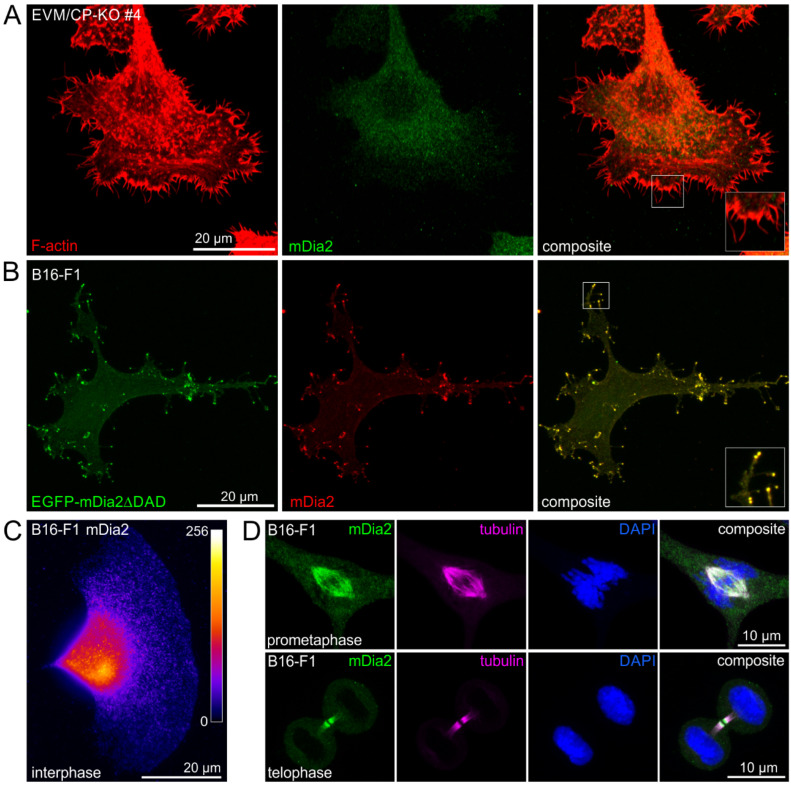
Endogenous mDia2 is not detectable at the tips of filopodia and lamellipodia but accumulates prominently together with tubulin at the mitotic spindle and midbody of dividing B16-F1 cells. (**A**) Representative image of an EVM/CP-KO cell stained for endogenous mDia2 and F-actin. Note the lack of mDia2 enrichment at filopodia tips. Inset, enlarged image of boxed region. Scale bar, 20 µm. (**B**) Ectopically expressed EGFP-mDia2∆DAD triggers filopodia formation in B61-F1 cells, with active mDia2 localizing to the distal tips of filopodia. Remarkably, even at low expression levels, ectopically expressed EGFP-mDia2∆DAD was effectively recognized by the anti-mDia2 antibody. Inset, enlarged image of boxed region. (**C**) Heat map of endogenous mDia2 signal intensity in a representative B16-F1 cell during interphase, forming a prominent lamellipodium. Note the lack of mDia2 enrichment at the lamellipodium tip. (**D**) Representative images of B16-F1 cells stained for endogenous mDia2, tubulin and DNA with DAPI in prometaphase (**upper panel**) and telophase (**lower panel**).

**Figure 7 cells-12-00890-f007:**
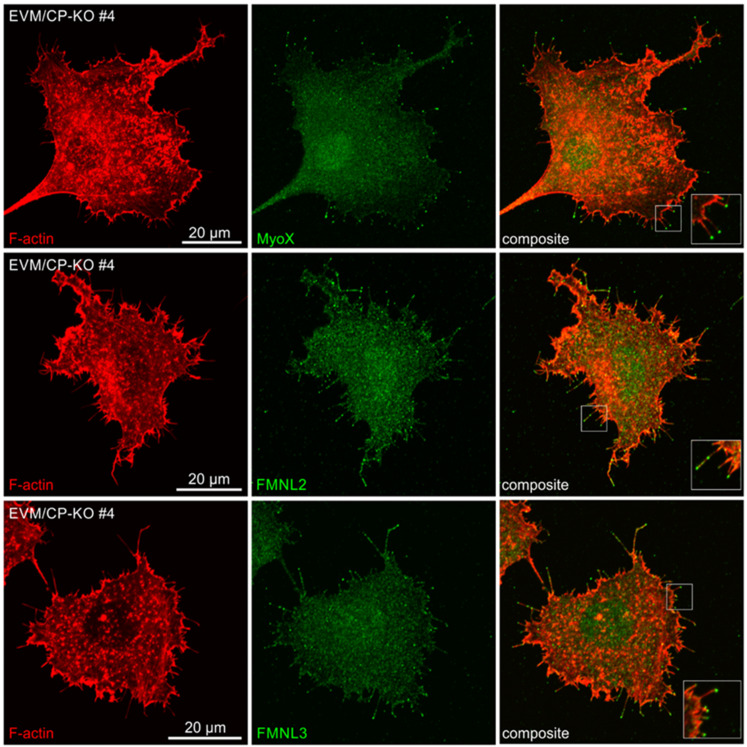
Endogenous MyoX, FMNL2 and FMNL3 accumulate at the tips of filopodia in EVM/CP-KO cells. Representative EVM/CP-KO cells stained for MyoX and F-actin (**upper panel**), for FMNL2 and F-actin (**middle panel**) and for FMNL3 and F-actin (**lower panel**). Insets, enlarged images of boxed regions.

**Figure 8 cells-12-00890-f008:**
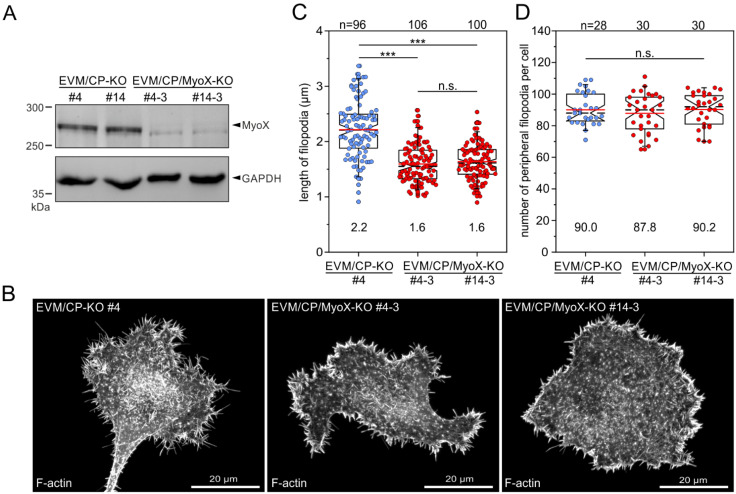
Loss of MyoX in EVM/CP-KO cells leads to additional reduction in filopodia length. (**A**) Immunoblot confirming elimination of MyoX in two independent EVM/CP/MyoX-KO mutants (clones #4-3 and #14-3). Loading control: GAPDH. (**B**) Morphologies of representative EVM/CP-KO and EVM/CP/MyoX-KO cells migrating on laminin and stained for the F-actin cytoskeleton with phalloidin. (**C**) Comparison of filopodium length in EVM/CP-KO versus EVM/CP/MyoX-KO cells. Note the significantly shorter filopodia in EVM/CP/MyoX-KO mutants. (**D**) Quantification of the number of peripheral filopodia in EVM/CP-KO and EVM/CP/MyoX-KO cells. (**C**,**D**) The boxes in box plots indicate 50% (25–75%) and whiskers (5–95%) of all measurements, with dashed black lines depicting the medians and arithmetic means highlighted in red. (**C**) Non-parametric Kruskal–Wallis test and Dunn’s Multiple Comparison test and (**D**) one-way ANOVA and Tukey Multiple Comparison test were used to reveal statistically significant differences between datasets. *** *p* ≤ 0.001; n.s.: not significant. n: number of peripheral filopodia (**C**) or cells (**D**) analyzed from three independent experiments.

**Figure 9 cells-12-00890-f009:**
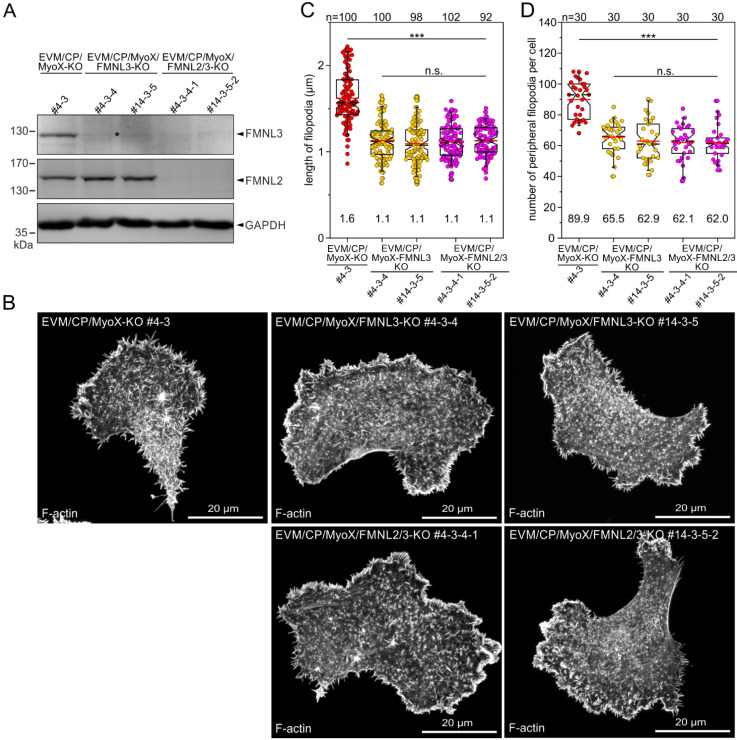
Additional inactivation of FMNL-family formins in EVM/CP/MyoX-KO cells affected not only the length but also the number of formed filopodia. (**A**) Immunoblot confirming the elimination of FMNL3 or FMNL2 and -3 in two independent EVM/CP/MyoX/FMNL3 or EVM/CP/MyoX/FMNL2/3 mutants (clones #4-3-4 and #14-3-5 or #4-3-4-1 and #14-3-5-2, respectively). The asterisk indicates a nonspecific band, as shown in more detail in Appendix A. Loading control: GAPDH. (**B**) Representative images of mutant cells indicated migrating on laminin and stained for the F-actin cytoskeleton with phalloidin. (**C**) Comparison of filopodium length in respective cell line. (**D**) Quantification of the number of peripheral filopodia in mutants as indicated. (**C**,**D**) The boxes in box plots indicate 50% (25–75%) and whiskers (5–95%) of all measurements, with dashed black lines depicting the medians and arithmetic means highlighted in red. (**C**) Non-parametric Kruskal–Wallis test and Dunn’s Multiple Comparison test and (**D**) one-way ANOVA and Tukey Multiple Comparison test were used to reveal statistically significant differences between datasets. *** *p* ≤ 0.001; n.s.: not significant. n: number of filopodia (**C**) or cells (**D**) analyzed from three independent experiments.

**Table 1 cells-12-00890-t001:** Comparison of length and number of peripheral filopodia in analyzed mutant cell lines. Values indicate mean ± SD of pooled data from independent clones of respective genotype. n: number filopodia ^1^ or cells ^2^ analyzed.

Mutant Cell Lines	Filopodia Length (µm)	Normalized to CP-KO (%)	n ^1^	Number of Filopodia/Cell	Normalized to CP-KO (%)	n ^2^
CP-KO	2.8 ± 0.7	100.0 ± 25.6	208	89.9 ± 14.3	100.0 ± 15.9	59
EVM/CP-KO	2.2 ± 0.5	78.6 ± 24.5	204	90.1 ± 14.2	100.1 ± 15.7	54
EVM/CP/MyoX-KO	1.6 ± 0.3	57.7 ± 20.6	206	89.0 ± 11.4	99.0 ± 12.8	60
EVM/CP/FMNL3-KO	1.1 ± 0.2	39.7 ± 20.7	198	64.2 ± 11.7	71.4 ± 18.3	60
EVM/CP/MyoX/FMNL2/3-KO	1.1 ± 0.9	40.0 ± 16.8	194	62.0 ± 10.8	69.0 ± 17.3	60

## Data Availability

All study data are included in this article. Constructs and macros presented in this study are available from the corresponding author upon reasonable request.

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
