# Peer review of "Unleashed Actin Assembly in Capping Protein-Deficient B16-F1 Cells Enables Identification of Multiple Factors Contributing to Filopodium Formation"

_cells, 2023, doi:10.3390/cells12060890_

Round 1
Reviewer 1 Report
The manuscript by Hein et al. describes the phenotype of mutant cell lines lacking capping protein and combinations of compound mutants, mainly Ena/VASP. In summary this is technically excellent work and the data are presented in a very clear manner. In terms of content I really have very little to criticise.
I would have only few minor points, the authors might consider in a final version:
lane 21: the wording long filopodia gives the wrong impression. What is a long filopodia versus a spike is a purely arbitrary men made parameter. I suggest to delete ‘long’. After all Ena/VASP does affect at least the length of filopodia, but not the frequency of initiation.
lane 98: this highlights a general issue which leads to unnecessary irritation in the field. Cells are simply different and data from one to the other cannot be transferred 1:1. I suggest to discuss/highlight the issue of ‘cell specificity’ in the actin field in the discussion section.
lane 597-600: FACS analysis is a powerful method to quantitate F-actin after phalloidin staining, however when interpreting the results one must not forget that the FACS results show a relative increase in F-actin NOT an absolute. For this one would have to determine the F/G-actin ratios via biochemical fractionation of TX-insoluble/soluble material. Therefore it is not surprising that the authors see differences to previous papers that used different methods. If the authors would perform a classical ‘McRobbie’ assay on their cells the differences would not be 5-fold more F-actin, my prediction.
Reviewer 2 Report
The paper wants to unleash new evidence-based science about driving factors responsible for filopodia formation. Through carefully crafted experiments involving among other many single clonal cell lines (which is by itself hard enough to do) the authors have clearly succeeded in demonstrating new and sometimes unexpected findings. From this work the entire scientific community can certainly benefit. Having said so, I nevertheless provide certain comments/questions to further expand the work clarity, understanding as well scientific integrity for broader community.
Major comments:
Since it was quite clearly demonstrated that entire process is cell line (sometimes via surface) depended the authors should provide certain scientific justification on cell line(s) choice, decision to use animal and not human cell line altogether and so one.
Multiple factors were included however few important are missing. For example, from myosin classes only Myosin X was examined however there are more myosins critically involved into the process like Myosin 19 and Myosin 2A. I do not expect more experiments however certain discussion should be provided.
Complete inhibition of filopodia formation was clearly not achieved despite big amount of work and intention provided. I wonder about the more clearly discussed reasons about this fundamental outcome of the study. Could be such “unexpected” results a consequence of better method(s) sensitivity used in this study in comparison to other literature? Or the use of different cell lines/surfaces? I believe a clearer statement/discussion is needed in this regard.
Image processing is appropriate only when it is applied equally across the entire image and is applied equally to controls. Excessive manipulations, such as processing to emphasize/diminish one region in the image at the expense of others (e.g. through the use of a biased choice of threshold settings), is usually inappropriate, as is emphasizing experimental data relative to the control. Thus, described manual correction (Line 264-265) of only part(s) of the image (manual removal of regions with high background around the nucleus), and not image as a whole is a tricky business. A full image sequence depicting each individual step of image processing should be provided in the SI depicting that no important feature is added or removed or unproportionally enhanced or diminished from the original image. The authors should consider depositing raw images/movies on suitable depositories (e.g., Zenodo, etc.).
In Figure captions the name of the cell line (B16-F1, etc.) should be always somehow included (e.g., in first sentence).
Line 163: if laboratory animal was indeed used in the study an approval number for local ethical committee for experiments on animal is missing!
Minor comments:
16: filopodia were nicely described however “ruffling” not so much. Would be nice to spend a word or two also here.
17: essential in terms of sufficient?
Introduction
36: dendritic actin: short description needed
65: some reference is needed at the end of such statement (sentence).
77: SF: state clear full name before first use (perhaps in Figure captions also)
95: CPβ: not clear from where β is coming from and why
97: not clear what was not observed
98: reference needed for statement till “MVD7 fibroblast”
98-99: is this hypothesis authors work, or it was stated before in published literature? If later the reference is needed.
M&M
103: Would be nice if any sequences of all those constructs and plasmids be deposited in some appropriate repositories (International Nucleotide Sequence Collaboration (INSDC, https://www.insdc.org/) or Genome Sequence Archive (GSA, https://ngdc.cncb.ac.cn/gsa/).
122: how long after seeding, the cells were transfected?
124: at least DNA:JetPrime ratio should be provided, and the transfection period used (4h, overnight, etc.). What was approximate average transfection efficiency reached? This can give us an extra clue about representativeness of results.
144: short description of “pre-conditioned medium” missing
148: at least basal purification buffers composition should be stated
161: a scientific reason should be provided why antibodies were generated via animal immunization and not by using modern animal-free techniques (antibodies developed using in vitro recombinant libraries and affinity reagents)
164-180: a dilution solution/buffer should be stated (also in the rest of the paper as in line 215)
184: 100 % confluency or?
185: an approximate volume of RIPA buffer vs specific number of cells (or cell area) should be given
190: some reasoning should be provided why SDS concentration was increased step wise.
195: a device used for blotting should be stated (with program or voltage used)
199: define (approximate) “extensive washing”
202: concentration of NaHCO3 missing
212: special section dedicated to chemicals should be added (with manufacturer, serial number, or similar provided for chemicals used, like gelatin, BSA, etc.)
220: exposure time, gain, illumination/laser power, any optical density filter used, image bit depth…missing for all image acquisitions (also for time laps)
225: short description of coating process should be stated
232: define “low density” in terms of number of cells / 35 mm dish
233: cell media conditions (e.g., starvation) it is known to be powerful filopodia initiator. Was this controlled, considered, discussed?
234: was illumination OFF (using e.g., light shutter) during the individual image acquisition?
238: time laps movies were generated in the study but non provided as SI movies. Some representative movies for each condition should be provided.
239: ImageJ / MtrackJ references should be provided
262: macro code should be provided in appendix
264: background subtraction was done on images with what image bit depth? This is important since relation between ball radius and objects is different on 8- or 16-bit images.
Results:
289: CapZb -> CapZβ?
297: Can some of the microspikes be pointed/marked somehow (arrows, etc)?
296-301: The images of Figure 1B should be noted/referred more individually, perhaps as Fig 1B (left, middle and right) with clear context in the text. Also, each image should have “F-actin” in bottom left edge.
305: Would be possible to generate similar composite of control cell?
311: mean+SD in both cases? Please state.
319-320: list some of “defect” and “effect”
327: Figure 1C: here CapZ-alpha seems to be present in one band (isoform?) while there are two bands (isoforms?) in the rest of the immunoblots. Can this have any effect on presented results?
331: The start of description of box plot is a bit out of context. Please make it more related to Figure 1E. For the all box-plots please make the lines thicker (e.g. marking medians…), also for the sake of consistency, make arithmetic means always in blue color.
343: define approximate “saturating conditions” in terms of [phalloidin] vs cell area or number. Was [phalloidin] the same for all conditions?
349: explanation is needed why only actin assembly (F-actin) is detected with this assay and not total actin content (related to Figure 2C).
357: more “polymerizable monomeric” actin?
359: Figure 2 B/D -> why are intensities of two controls so different? 2-3-fold? It should be somehow noted/explained.
382: reference which links SF to adhesiveness should be added
433: Thus, one can conclude that B16-F1 were also poorly attached…Is that the case? Should be discussed/noted.
437: Figure4: would be beneficial to see more representative cells (in supplemental information), perhaps uncropped images of lower magnification (valid for any Figure where only one representative cell per condition is shown).
507: Are mDia2 implications supposed to be valid for the cell lines used in the study or for cells in general? Should be specified.
582: Figure 9: “*” how can one know / be sure that this band is unspecific? Certain explanation needed.
Discussion
611-612: perhaps I missed something altogether, but if we have 2x stronger expression and 4x higher F-actin content, how can such cell does not reach physiological actin concentration? Would not be in total even more [actin] in CP-KO cells than in wild type?
727: Table 1: is “CP-“equal to “CP-KO”? If so, the use of the same consistent expression is preferable.
Original blot images:
Original blots should have membrane edges visible. kDa on marker should be stated.
Supplemental figures:
Bar in µm missing in some of the images
Reviewer 3 Report
The dynamics of the actin cytoskeleton and generation of protrusive structures is driven by the complex interplay and balance of activities by actin regulators. The heterodimeric capping proteins (CP) play especially important roles in limiting the growth of actin filaments. Their activity is blocked by two polymerases implicated in filopodia formation that build parallel actin bundles, Ena/VASP and formins. Depletion of CP has been found to cause the dramatic overgrowth of filopodia, thin actin-filled membrane protrusions. Hein et al test the hypothesis that Ena/VASP and/or filopodial formins are required for the formation of excess filopodia when actin polymerization goes unchecked in the absence of CP. Unexpectedly, elimination of CP-beta from NIH 3T3 cells (via CRISPR genome editing) results in increased levels of F-actin and a dramatic increase of stress fibers (SFs) accompanied by formation of increased numbers of focal adhesions (FAs). Increased filopodia are only observed if the cells are not well-adhered to the substrate. These observations are quite surprising and the basis for this notable effect is unclear. On the other hand, loss of expression of the CP-beta subunit in the B16-F1 mouse melanoma line also results in a loss of its binding partner CP-alpha (as seen for NIH 3T3 cells), the appearance of numerous filopodia and a significant decrease in cell migration. Subsequent deletion of Ena/VASP family members (all three) then the filopodial myosin MyoX (EVM/CP/MyoX-KO) in the CP-beta mutant line does not abrogate filopodia formation but does result in somewhat shorter filopodia. Loss of CP-beta, Ena/VASP proteins and the two known filopodial formins, FMNL2 and FMNL3 (EVM/CP/FMNL2/3-KO) causes a 30% decrease in filopodia number and further shortening of filopodia. These data show that formation of wild type levels of filopodia following loss of CP activity requires other actin regulators that remain to be identified.
The results are striking and raise interesting questions about how cells lacking several key filopodial proteins can still make filopodia or amplify formation of FAs and SFs. The main problem is that the authors present these two interesting findings, neither of which can be clearly explained at present. The results do certainly highlight the adaptability of the actin cytoskeleton and how loss of a key controller of actin filament growth plays a pivotal role in maintaining cytoskeletal balance, but definitive answers to the questions raised here would appear to await future investigation.
One major issue is that the work is presented as if loss of one (or more) protein has a straightforward effect, one piece of machine is removed and nothing else is altered. Cells have a remarkable ability to adapt by up-regulating or down-regulating isoforms or proteins that have related activities in order to carry on. The genetic background can also make a big difference, as clearly seen in this work. The authors do discuss the possibility (or not) that other members of the CP and formin families or even regulators of the Wave Regulatory Complex as candidates (or not) could be responsible for filopodia formation in the KO lines or SF and FA formation in NIH 3T3 cells but they do not go any further than that. The focus is largely on considering proteins that must reside at filopodia tips but it is known that overexpression of the actin bundler fascin can promote filopodia formation so the role of proteins with this type of activity should be considered. Until a more extensive study can be carried out, a few additional, focussed experiments to test the possible upregulation or re-localization of some select additional candidates in the B16-F1 KO lines could provide useful information to help readers make better sense of the results presented here. For example, are FMNL2/3 upregulated in the EVM/CP/MyoX-KO line or is MyoX expression altered in the EVM/CP/FMNL2/3-KO line?
Comments
The authors perform TIDE analysis to characterize the deletions in their mutant lines. The results of that characterization should be provided for any of the mutants generated for this study so readers can know the nature of the deletion and how it disrupts the coding sequence of the targeted gene.
Total actin levels in the CP-beta KO are quantified by western blotting (Fig 2E, F). It is unclear if the detection system used for western analysis (phosphatase-conjugated antibodies and band detection by BCIP) provides the authors with good linearity. Over- or under-development of a colorimetric signal can give misleading results. How were the blot images captured and how did the authors establish that the signals are in the linear range of their method and imaging system?
A critical measurement here is counting filopodia numbers and lengths (Figs 5, 8, 9 and Table 1) that must be challenging given the large numbers of filopodia produced in the KO cells (based on the few images shown). The method for doing this is not at all described - how were individual filopodia identified, counted and measured reliably? What is the background level of filopodia in the parental B16-F1 cell line? The B16-F1 cells can either have a polarized morphology or not - did all of the KO cells have various shapes or were they rather uniform? Were all cells analyzed regardless of morphology? Presumably so, but this information should be provided.
Filopodia length is thought to be linked to stabilization by adhesions formed with the substrate - do the multiple KO (EVM/CP or EVM/CP/MyoX) cells exhibit any changes in the formation of adhesions or spreading?
There appears to be a faint band running quite close to the molecular weight of full-length MyoX in the blot of the EVM/CP/MyoX-KO lines shown in Fig. 8A. The analysis presented here is consistent with the lines being deleted for MyoX, but the band does raise the question of how was the MyoX anti-sera validated? Also, can the authors firmly rule out the possibility that their lines have either some residual expression of MyoX or that they are not fully clonal?
Similarly, there is a faint band in the blot of the FMNL3 knock-outs that is also running close to or at the size of the FMNL3 protein (Fig 9A). The authors state in the legend that this is a non-specific band but they do not provide any supporting evidence for this assertion. The phenotype of the 6x or 7x KOs are consistent with loss of protein function but the presence of a band raises the questions about the homogeneity of the population full effectiveness of the KO (i.e a small level of remaining expression).
Minor comments
Please provide a citation supporting the statement that a loss of MyoX abolishes microspikes (pg 2, line 92).
Also, the authors indicate that filopodia numbers are decreased by 50% following knockdown of MyoX in Hela cells (pg 2, line 92). In looking at the cited paper (ref 38 - Bohil et al) it would appear that the decrease is more substantial than that, it looks more like a 75% decrease in filopodia number.
A 'very recent study' showing a severe impairment in lamellipodium protrusion in CP depleted B16-F1 cells is referred to in the Discussion (pg 19, line 618-620) but the cited reference 11 is a paper from 2002 about WAVE1 and actin nucleation. Is that really the correct citation?
The authors discuss the relationship between membrane tension and lamellipodial extension (pg 12, lines 424-427) and cite the excellent review on Ena/VASP by Faix & Rottner (2022) but there does not appear to be much of a discussion on this point in the article. It would be helpful if a more relevant citation could be provided.
Ref 45 is incomplete - it is missing the name of the book series, book and page numbers.
The authors refer to a custom macro that was employed for the analysis of focal adhesions (page 6, line 262) but there is no indication of how this macro might be available to interested researchers.
page 20, line 637 "attempting to speculate…" should likely be 'tempting' to speculate
Round 2
Reviewer 2 Report
I here conclude that the authors addressed the comments very well and appropriately modified the manuscript which is now suitable for publication.
I would like to, nevertheless, encourage them to include in the final version (perhaps as supplemental information) the step-by-step image analysis and (some of) evidence which support unspecific band judgment (marked by asterisk). As work was nicely done for review purpose, I believe it deserves to be available also to future readers of the manuscript.
Author Response
Done. We have added a new Supplemental Figure S8 to support unspecific band judgment (marked by asterisk in main Figure 9).
Reviewer 3 Report
The authors have nicely addressed all of the comments from the original review.
One minor addition needed is the description of the method for detecting the enhanced chemiluminescence signal (film or type of imager).
Author Response
Done. We have amended the method section (2.9) now stating: "...Primary actin antibodies in these immunoblots were visualized by enhanced chemiluminescence using the ChemiDoc MP Imaging System driven by Image Lab software (Biorad, Hercules, CA, USA). "